# A new approach to continuous monitoring of carbon use efficiency and biosynthesis in soil microbes from measurement of CO2 and O2

Kyle E. Smart[1], Daniel O. Breecker[2], Christopher B. Blackwood[3,4,5], and Timothy M. Gallagher[1]

[1]Department of Earth Sciences, Kent State University, Kent, OH, 44242, USA.
[2]Department of Geological Sciences, University of Texas at Austin, Austin, TX, 78712, USA.
[3]Department of Plant, Soil and Microbial Sciences, Michigan State University, East Lansing, MI, 48824, USA.
[4]Department of Plant Biology, Michigan State University, East Lansing, MI, 48824, USA.
[5]Ecology, Evolution, and Behaviour Program, Michigan State University, East Lansing, MI, 48824, USA.

*Correspondence to*: Kyle E. Smart (ksmart4@kent.edu)

**Abstract**

Soils comprise the largest terrestrial carbon pool. Therefore, understanding processes that control soil carbon stabilization and release is vital to improving our understanding of the global carbon cycle. Heterotrophic respiration is the main pathway by which soil organic carbon is returned to the atmosphere, however not all carbon utilized by heterotrophs shares this fate, as some portion is retained in the soil as biomass and biosynthesized extracellular compounds. The fraction of carbon consumed by microbes that is used for biomass growth (the carbon use efficiency or CUE) is an important variable controlling soil carbon stocks but is difficult to measure. Here we show that CUE can be continuously monitored in laboratory glucose-amended soil incubations by measuring $CO_2$ and $O_2$ gas concentrations, allowing instantaneous estimates of microbial biomass growth. We derive a theoretical relationship between the respiratory quotient (RQ), the ratio of carbon dioxide produced to oxygen consumed during respiration, and CUE that recognizes the influence of both substrate and biosynthesized product oxidation states on RQ. Assuming the biosynthesized product has the stoichiometry of an average microbe, and that the substrate is primarily the glucose used for amendment, we measure RQ and use our theoretical relationship to calculate CUE, and from that, biomass production. Extractions of microbial biomass carbon at the end of the experiments reveal minimal net increases in standing biomass across all amended treatments, suggesting that much of this newly produced biomass is likely converted to necromass as substrate availability declines and this results in a net storage of new soil organic matter. Carbon budgets compiled from measurements of relevant pools account for the amended carbon and suggest that with larger carbon amendments, increases in C:N ratios lead to increases in the relative portion of the amendment acutely lost from the soil. These findings demonstrate that soil RQ values may be used to monitor changes in CUE and that studies which monitor soil RQ values should consider CUE as a key factor when changes in RQ are observed, for instance, with changing environmental conditions or changes in production of plant derived compounds. This new approach may be leveraged to provide information on the storage of soil organic matter. These findings demonstrate how measurements of soil RQ may be leveraged to understand soil carbon transformations, specifically the fate of fresh carbon inputs.

## 1 Introduction

Soils represent one of the largest pools of carbon on the Earth's surface, with 1477 Gt of carbon stored as soil organic matter (Scharlemann et al., 2014). The makeup of this pool can change dynamically as organic carbon is added through litter and root inputs, transformed by soil biogeochemical processes, and ultimately released back to the atmosphere via respiration (Dynarski et al., 2020; Kögel-Knabner, 2002; McDaniel et al., 2014; Paul, 2016a). These exchanges of carbon are of particular importance, because as climate conditions continue to change and natural ecosystems exist in a state of increasing disequilibrium from antecedent conditions, it is difficult to predict the rates at which soils will accumulate or lose carbon. The processes that control soil carbon cycling are crucial to understand, not only in the context of global climate (Scharlemann et al., 2014), but also because soil organic carbon impacts soil fertility directly by providing essential nutrients and compounds for plants and microbes and indirectly by affecting soil physicochemical properties like wettability and drainage (Gaiser and Stahr, 2013). Therefore, improving our understanding of these processes may also better our efforts of conserving soil organic carbon in the context of global food security.

To understand if soils are experiencing a net gain or loss of carbon, it is necessary to first examine the interplay of biosynthesis and respiration (Adingo et al., 2021; Blagodatskaya et al., 2014; Geyer et al., 2016; Manzoni et al., 2018; Sinsabaugh et al., 2013). Accurately quantifying heterotrophic respiration is critical because it is the main mechanism by which carbon is released from soils (Landsberg and Gower, 1997; Mukul et al., 2020; Walker et al., 2018). Microbes consume soil organic matter not only as a source of energy via respiration, but also as a source of reduced carbon compounds for biosynthesis (Adingo et al., 2021; Schimel and Weintraub, 2003; Schimel and Schaeffer, 2012; Sinsabaugh et al., 2013). The proportion of carbon consumed by microbes that is retained in biomass, rather than respired, is known as the Carbon-Use Efficiency (CUE). Biosynthesis of microbial biomass and extracellular compounds is important to constrain because it is thought to be an important pathway for long-term stabilization of organic carbon within soils (Cotrufo et al., 2013, 2015a; Miltner et al., 2012; Wieder et al., 2014). As soil microbes take up new organic carbon from fresh plant litter or other soil organic matter, CUE is the first crucial step in determining the fate of the consumed carbon (Kästner et al., 2021; Kindler et al., 2009; Liang et al., 2019; Miltner et al., 2011; Paul, 2016b; Wang et al., 2021). After the stimulation of growth, newly produced microbial biomass is converted to necromass, as cell death occurs on the order of hours to days (Buckeridge et al., 2020). This necromass contains an abundance of molecules which may be further metabolized or recycled for molecular maintenance. However, not all this necromass is likely to be immediately accessible, due to factors including physical occlusion, chemical lability vs. recalcitrance, stabilization onto mineral surfaces, or continued supply of more desirable compounds (Buckeridge et al., 2020, 2022; Cotrufo et al., 2015b; Hu et al., 2023; Kästner et al., 2021; Liang et al., 2019; Lützow et al., 2006; Paul, 2016b; Wang

et al., 2021). Regardless of the exact mechanism, many studies have shown that microbial necromass residues should be considered an important pool through which organic matter cycles and stabilizes in soils.

The concept of CUE can be applied at different spatial and temporal scales, depending on the question of interest (Adingo et al., 2021; Geyer et al., 2016, 2019). For example, it may be useful to consider the CUE of individual microbial community members when studying ecological processes like competition or response to changes in environmental conditions. The CUE
of the community as a whole may also be estimated when studying factors like ecosystem oxidation state (Geyer et al., 2019; Sinsabaugh et al., 2013). There is also debate as to whether CUE is an inherent species-specific value, and constant, or if CUE is a variable that can change over time given the needs of the microbes and the environmental conditions (Adingo et al., 2021; Geyer et al., 2016; Manzoni et al., 2012, 2018; Sinsabaugh et al., 2013). Regardless, CUE is crucial for understanding soil organic carbon stability because at low values, soil carbon is 'burned off' where at high values it is efficiently recycled.
Unfortunately, CUE has been difficult to measure and nearly impossible to monitor continuously.

An emerging approach that can be used to study soil metabolisms and other soil processes is known as Respiratory Quotient (RQ), which is the ratio of $CO_2$ produced to the $O_2$ consumed during respiration (Dilly, 2001, 2003). The study of RQ can potentially provide insight into the substrate being metabolized because the stoichiometry of the compound should determine the reaction stoichiometry during aerobic respiration (Masiello et al., 2008). For example, respiration of compounds like sugars
and other carbohydrates are predicted to produce an RQ of 1.0, lipids are predicted to have RQ values around 0.7, and most organic acids around 1.3 (Hicks Pries et al., 2020; Hilman et al., 2022; Masiello et al., 2008). While some studies report RQ values that resemble substrate-based predictions, other studies observed systematic deviations that were linked to non-metabolic processes which can affect soil $CO_2$ and $O_2$ fluxes, such as different diffusion constants of $CO_2$ and $O_2$ which can be accounted for and presented as Apparent Respiratory Quotient (ARQ) (Angert et al., 2015). Other non-metabolic processes
known to affect measured RQ values include: calcite dissolution/precipitation which can cause a transient decoupling of these two gases, and oxidation of reduced metal species which can cause additional draw down of oxygen (Angert et al., 2015; Bergel et al., 2017; Gallagher and Breecker, 2020; Hicks Pries et al., 2020; Hodges et al., 2019; Sánchez-Cañete et al., 2018). Additionally, the presence of anaerobic respiration can complicate measurements of RQ as microbes utilize alternative terminal electron acceptors to carry out their metabolism which contributes additional $CO_2$ without any corresponding consumption of
$O_2$. Such processes are particularly important in field studies of water-logged soils and potentially when intact soil aggregates allow for the persistence of anaerobic microsites in otherwise well-oxygenated soils (Keiluweit et al., 2018; Tiedje et al., 1984). Although in the latter case, the impact of anaerobic pathways on RQ values will be minimal if aerobic respiration rates are orders of magnitude larger than anaerobic respiration rates.

The potential effect of microbial CUE on soil RQ values has received less attention to date, although Dilly (2001) suggested
that incorporation of available substrates into microbial biomass could explain initial RQ values >1 observed during the initial stimulation period in soils amended with glucose. If microbial biosynthesis causes divergence of observed RQ values from

expectations derived from substrate stoichiometry alone, then by examining the effects of CUE on RQ we may enable indirect monitoring of biosynthesis through the measurement of RQ. In order to examine the utility of RQ as a CUE-indicator, we designed incubation experiments in which glucose was added as a substrate to induce respiration, $CO_2$ and $O_2$ in the incubation vessel headspace were measured every 2 h and biomass was measured by the chloroform-fumigation extraction method once respiration rates declined to baseline values. We further explore the implications of the biosynthesis we infer from the measurements in the context of the fate of soil organic carbon transformations.

## 2 Connecting Carbon Use Efficiency and Respiratory Quotient

When substrate is converted entirely to $CO_2$ and yields no net biomass production, carbon use efficiency is zero and does not influence RQ. When CUE is non-zero, RQ values are driven by the difference between the oxidation states of carbon in substrate and reaction product (i.e., between the molecule consumed and the molecule produced through anabolism). To understand how biosynthetic processes influence RQ, we must describe how changing CUE will influence this stoichiometry by considering the production of microbial biomass as a key reaction product. Using a mass balance approach, we can explore the relationship between RQ and CUE in the reaction:

$$A\ C_6H_{12}O_6 +\ B\ O_2\ +\ F\ NO_3^-\ =\ C\ CO_2\ +\ D\ H_2O\ +\ E\ C_1H_{1.8}O_{0.5}N_{0.2} \tag{1}$$

where $C_1H_{1.8}O_{0.5}N_{0.2}$ is a representative microbial biomass stoichiometry (Roels, 1980) normalized per mole of carbon, and the letters A - F serve as coefficients. Due to its relative importance in microbial makeup, nitrogen was included in the calculations. We chose to use nitrate as the nitrogen bearing substrate due to its impact on RQ values by its redox state and widespread occurrence in soils. A derived theoretical relationship between RQ and CUE, following Eq. 1 is shown below (Fig.1), and is further applied to experimental data to address our research questions. Derivation of the relationship between RQ and CUE occurred as follows:

Define elementally specific mass balance expressions.

Carbon: $6A = C + E$ or $E = 6A - C$ (2)

Hydrogen: $12A = 2D + 1.8E$ (3)

Oxygen: $6A + 2B + 3F = 2C + D + 0.5E$ (4)

Nitrogen: $F = 0.2E$ (5)

Define CUE and RQ as a function of coefficients.

$CUE = E / (E + C)$ (6)

$RQ = C/B$ (7)

Start with Eq.4 and substitute Eq.5 to remove F.

$$6A + 2B + (0.2E) = 2C + D + 0.5E \tag{8}$$


Next substitute Eq.3 (solved for D).

$$6A + 2B + 0.2E = 2C + (6A - 0.9E) + 0.5E \tag{9}$$

Simplify

$2B + E = 2C$ (10)

Substitute Eq.6 (solved for E).

$$2B + (CUE(C) / (1 - CUE)) = 2C \tag{11}$$

Solve for RQ as a function of CUE, and CUE as a function of RQ

$$RQ = (2 - 2CUE) / (2 - 3CUE) \text{ or } CUE = (2RQ - 2) / (3RQ - 2) \tag{12}$$

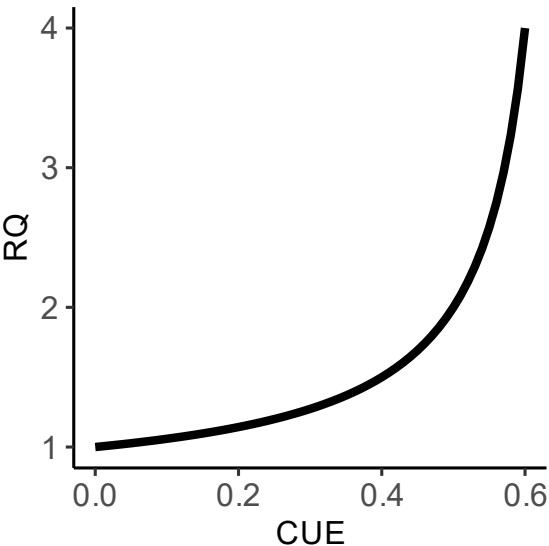

Figure 1: The calculated relationship between carbon use efficiency and respiratory quotient for Eq.1. This modeled relationship shows that as CUE increases, RQ values will also increase, which may seem counter-intuitive at first, given that an increase in CUE would cause a net decrease in $CO_2$ production, all else being equal. However, the concurrent $O_2$ consumption decreases more substantially as uptake of $NO_3^-$ increases, which in turn results in RQ values increasing. The slope of the modelled relationship shows that RQ increases rapidly as CUE values approach 60%. We limited our calculations

to a maximum of 60% CUE, as this is referenced as a theoretical thermodynamic limit for microbial metabolisms (Sinsabaugh et al., 2013).

## 3 Materials and Methods

This study consists of two soil incubations designed to investigate the effects of labile substrate (glucose) amendment on RQ values at high temporal resolution, and to evaluate the effects of CUE on RQ. Control samples (addition of Type 1 deionized
water, Millipore Milli-Q, to the soil) were incubated and measured for comparison. Treatment samples involved amendment with various masses of glucose (100 mg, 200 mg, 500 mg, or 1.0 g). Each of the two incubations consisted of two control samples, and six treatment samples. All incubated samples contained 20 g of soil. RQ was determined by monitoring the composition of headspace gas in the incubation vessels every 2 h for the duration of the incubations (262 h at longest).

The soils used in incubations were collected from a temperate deciduous forest in Portage County, Northeast Ohio. Soils in this location are designated as Chili Loam by the USDA Soil Survey. Soil collection was performed with a shovel, excavation included approximately the top four inches of the profile to include the Oi-horizon, and top 5 cm of the A horizon. Soil was then returned to the lab and homogenized. For purposes of incubation, the field moist soils were passed through a 2 mm sieve to remove large detritus and leaf litter and to break up large aggregates. Soils were then allowed to dry down, open to lab air,
for 2 weeks to encourage the depletion of any preexisting labile carbon and reduction of standing microbial biomass. Soil aliquots of 20 g (approximately 30 mL) were added to each incubation bottle. Glucose amendments were weighed and added to the soils as a fine solid powder, homogenized through physical mixing, and placed in 500 mL bottles. Once in the bottles, 10 mL of Type 1 deionized water (Millipore Milli-Q) was dripped evenly over the soils to encourage glucose dissolution before the bottles were capped and connected to the gas sampling apparatus. The rationale for glucose addition as a fine solid powder
prior to wetting was to prevent rapid uptake before the bottles could be capped and measurements could commence. The addition of water led to an average soil moisture of 36% by mass at the start of incubation. Incubations were carried out in an incubator held to constant temperature of 20 °C.

### 3.2 Automated Gas Sampling Apparatus

An automated gas sampling apparatus was constructed that allowed gas samples to be continuously collected and measured
from soil incubations every two hours. Soils were incubated in 500 mL glass bottles (PYREX[TM]) with 3 gas-tight tube ports in the lid (Duran® GL45). One port on each bottle was connected to a Calibrated Instruments (McHenry, USA) Cali-5-BondTM gas sampling bag, filled with an additional 300mL of $CO_2$-Free Air to give the incubation vessel a variable volume, which enabled gas samples to be collected and new gas to be added while maintaining atmospheric pressure. Bev-A-Line IV tubing connected the bottles through a second port in the lid to a central manifold block with solenoid valves. The third port
was closed off and was not used in this study. A Sable Systems (Las Vegas, USA) FOXBOX was used to measure high

precision $CO_2$ and $O_2$ gas concentrations. All sampled gas was dried using PermaPure (Lakewood, NJ, USA) Nafion™ Tubing, passing through a separate 500 mL bottle containing magnesium perchlorate, and held at partial vacuum, prior to measurement. The configuration of the sampling apparatus is depicted below in Fig. 2. From the central manifold system gas flow could be (1) closed, (2) directed from the bottles into the FOXBOX, or (3) directed from compressed gas cylinders into the bottles. The manifold system could also direct flow of the compressed cylinders directly to the FOXBOX.

The entire system was controlled by a programmable logic controller (PLC), which automatically opened and closed solenoid valves, directed the flow of gas through the system, and logged data from the FOXBOX. Every two hours a measurement sequence would begin whereby bottles were sequentially measured for 3.5 min at a flow rate of 50 mL min$^{-1}$ for a total sample of 175 mL of gas. To maintain high temporal resolution measurements (2 h), a maximum of eight individual samples could be incubated simultaneously. To account for any short-term drift in measured $O_2$ values, ambient air was automatically measured directly from the laboratory HVAC inlet vent, between sample measurements. Sampling from HVAC inlet vent was preferred over lab air because HVAC air is a mixture of air sources from throughout the building and would provide a more stable measurement of $O_2$, whereas lab air $O_2$ concentration may fluctuate more dramatically with changes in room occupancy or sampling exhaust. Additionally, gas cylinders were measured containing zero (CO2-Free Air) and calibration (5000 ppm CO2) gasses to account for long-term measurement reproducibility. Lastly, after sampling from each bottle, the 175 mL of gas removed for analysis was replaced with CO2-Free Air by directing cylinder flow through a needle valve and a mass-flow meter into the incubation vial-gas bag system. The resulting dilution of $CO_2$ and addition of $O_2$ within each bottle was accounted for when calculating moles of $CO_2$ produced and $O_2$ consumed between measurements. All aspects of this bottle incubation design have been carefully chosen in order to encourage aerobic respiration to be dominant and minimize the possibility for anaerobic respiration to take place, which would impact our measurements and confound our findings.

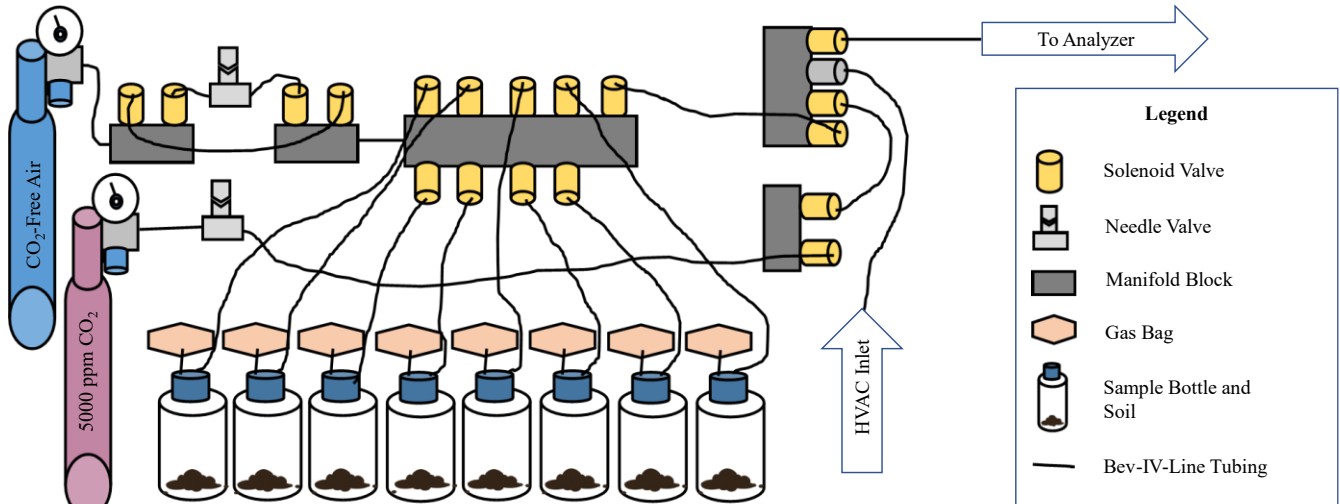

Figure 2: Experimental apparatus. This diagram displays the configuration of components used to construct the automated

gas sampling apparatus. Blue and pink gas cylinders on the left are labelled by type. All other components are identified in the legend. Arrows indicate "HVAC Inlet" used between sample measurements to separate measurement periods, and "To Analyzer" as the final outflow to the sample drier and FoxBox.

### 3.3 Microbial Biomass

Microbial biomass carbon was measured via the Chloroform Fumigation Extraction method following the methods of (McDaniel et al., 2014; Vance et al., 1987) on initial material at the start of the incubation and on the control and treatment samples at the end of the incubations. In short, duplicate subsamples (~5 g) were weighed out and one set were immediately extracted with 0.5M $K_2SO_4$, on a rotator table for 1 hour; these samples served as unfumigated water ($K_2SO_4$) extractable carbon. Next, the remaining samples were fumigated using ethanol-free chloroform (1mL) and capped for 24 h in a fume hood, then extracted with $K_2SO_4$; this set would serve as fumigated extractable carbon. All extracts were filtered through a Whatman #1 filter with a vacuum filtration apparatus immediately following extraction. Soil moisture measurements were carried out with the use of a drying oven and were determined gravimetrically on a third subsample of ~ 5 g of soil. Non particulate organic carbon was measured using a Shimadzu TOC-L Analyzer (Shimadzu Scientific Instruments Inc.) and reported in dissolved organic carbon (DOC) in mg $L^{-1}$. Dissolved organic carbon for both fumigated and unfumigated subsamples were used to calculate biomass carbon as Fumigated DOC – Unfumigated DOC = Biomass Associated DOC in mg C $g^{-1}$ dry soil. Final values are reported on a per bottle basis. A correction factor ($K_{ec}$= 0.45) was applied to account for the extraction efficiency of biomass carbon by chloroform, to convert Biomass Associated DOC to Biomass C (Vance et al., 1987). Salt Extractable carbon is presented as the unfumigated DOC and reported in mg C per bottle. Microbial biomass extraction was conducted on initial soil, on incubated control soil, and incubated amended soil. Incubated samples were harvested for biomass extraction immediately following decline in respiration stimulated by the amendment, and when measured RQ values drop below 1.0 for all replicates in each treatment group.

### 3.4 C:N Measurement

C:N values were determined for soil samples using a Costech Elemental Analyzer (EA) ECS 4010 configured with a CNH combustion column. In short, the dried subsamples used to collect soil moisture information as part of the CFE method were ground to a fine powder and weighed out into tin capsules. Reported values are given as the ratio of carbon to nitrogen in percent weight.

### 3.5 Data Analysis

Following the incubation, raw gas concentration data were processed in RStudio to quantify sample $CO_2$ and $O_2$ concentrations and apply a baseline correction. The baseline correction is done with a linear fit to HVAC air measurements made immediately preceding sample measurements. These HVAC measurements were corrected to 20.95% $O_2$. This correction is necessary to account for short-term drift on the fuel cell $O_2$ sensor, mostly caused through changes in temperature either by ambient

temperature or through heat dissipation within the instrument. Once the HVAC measurement corrections are established, the same correction is applied to sample measurement windows. Reported values of each sample are taken as the average value during the last 20 s (measurements are recorded every 2 s, 10 consecutive measurements are used) of the sampling window and an uncertainty is reported as the standard deviation. These drift-corrected data are then exported from RStudio into Excel for further processing. In Excel, measured $CO_2$ concentrations were corrected using a 2-point linear calibration curve produced

from measurements of $CO_2$-Free Air and 5000 ppm $CO_2$ gases. A mass balance approach was then used to calculate the moles of $CO_2$ produced (Fig. 3a) and $O_2$ consumed (Fig. 3b) during each 2 h incubation window, accounting for the dilution effect of replacing the sampled gas volume with 175 mL of CO2-free air after each analysis. With these data, RQ values for each 2 h interval were calculated (Fig. 3c). The variables of interest are saved in .csv files and imported to RStudio equipped with R version 4.2.2. Variables of interest include: time, $CO_2$ production rate, $O_2$ consumption rate, RQ, treatment, and replicate.

Periods of substrate induced respiration are defined here as being represented by an RQ $\geq 1.0$ and occurring during periods of elevated $CO_2$ production. CUE values were then calculated at each measurement of RQ using the relationship in Fig. 1, during the previously defined periods of substrate induced respiration. Following this, CUE and CO2 production rates were used to calculate moles of biomass carbon produced for each 2 h measurement interval. All variables, both measured and calculated, were then plotted. Packages employed in R include tidyverse, gridExtra, cowplot. and svglite. After analysis of the data, it was

determined that one of the eight bottles had an unnoticed leak during both incubation runs, so for these experiments only duplicate results are presented in Fig. 3, and these data from the problematic bottle were removed from presented averages in Fig. 4 and Fig. 5 (200 mg and 1000 mg incubations). Data from the first 2 h were not plotted in Fig. 3 and Fig. 4, as initial measurements produced a transient signal showing incredibly large $O_2$ consumption, which was likely the result of the bottles equilibrating with the new system. Presented values begin at 4 h.

**4 Results and Discussion**

**4.1 High Temporal Resolution RQ**

Respiration of glucose, and other simple carbohydrates, should produce a RQ value of 1.0, if CUE = 0  (Masiello et al., 2008). We observe RQs systematically greater than 1.0 post amendment, suggesting CUE > 0. Using mass balance calculations, we determined RQ values with a 2 h resolution (Fig. 3c), over the duration of 262 hours (10 days and 22 hours). Initial rates of

$CO_2$ production over the first 24 hours (Fig. 3a) show a similar overall trend regardless of amendment quantity, with all four amended treatments resulting in almost identical values. The rate of increase in $CO_2$ production initially appears to be inversely related to the amendment quantity, as the smaller amendment treatments begin to grow slightly faster than the larger amendments. Around 80 h of incubation, the $CO_2$ production rate of the 100 mg treatments peaked and declined over the remainder of the incubation. Peak $CO_2$ production for the 200, 500, and 1000 mg treatments occurred at 46-60, 78, and 84-92

hours, respectively. Notably, the 500 mg treatments reached comparable maximum $CO_2$ production rates with the 1000 mg treatments, suggesting that substrate availability alone is not a reliable predictor of yield in peak microbial respiration. One

possible explanation for this trend is slower dissolution of the glucose amendment in the 1000 mg treatment, which is supported by the data, and extended period of enhanced $CO_2$ production and thus greater cumulative $CO_2$ production in the 1000 mg treatment (Fig. 3a). Oxygen consumption rates displayed in Fig. 3b show a similar behavior to $CO_2$ production rates in Fig. 3a, apart from variability between timepoints and maximum values reached. Oxygen consumption rates occurred in a smoother, less erratic trend. Also, important to note is that in the control bottles oxygen consumption and carbon dioxide production rates did not change during the incubation period in any meaningful way.

Initially, at 4 h of incubation, RQ values across all treatments were noisy and ranged between ~0.3 - 1.5, probably related to error associated with determining RQ when respiration rates are small. From 4 h onward, RQ values in amended treatments start an overall ascent. After ~24 h of incubation, coinciding with an increase in $CO_2$ production and $O_2$ consumption, RQ values across most treatments are > 1.0. While the rates of gas exchange continue to climb, RQ values also increase. RQ values observed during peak respiration (~1.3-1.6) are similar across treatments. As the rates of $CO_2$ production begin to decline, RQ values also decline. Although treatment replicates are variable with respect to time, the overall trends are in good agreement. RQ is not shown for control samples because we observed no overall trend (i.e. no increase or decrease). We see that RQ values are dynamic at this temporal resolution, even during the period which should be dominated by substrate induced respiration, meaning that RQ values are not simply a direct result of the substrate being oxidized to produce $CO_2$.

Peak RQ values were observed during peak respiration and are similar to those observed in (Dilly, 2001), with RQ values ~ 1.5. Notably, all treatments measure ~1.5 despite an order of magnitude increase in glucose amendment. This suggests that biosynthetic processes are limited by the rate of synthesis of biomolecules perhaps with temperature or availability of other nutrients (eg. N or P) acting as controls. Also importantly, we see that the overall range in RQ values is quite large (0.3 – 1.9). These higher values could be explained through partially anabolic metabolism; however values below 1.0 likely indicate the use of some other substrate in which the carbon is more reduced. This other carbon substrate could be a form of less labile organic matter contained within the initial soil samples, or metabolites that were produced during the respiration of glucose.

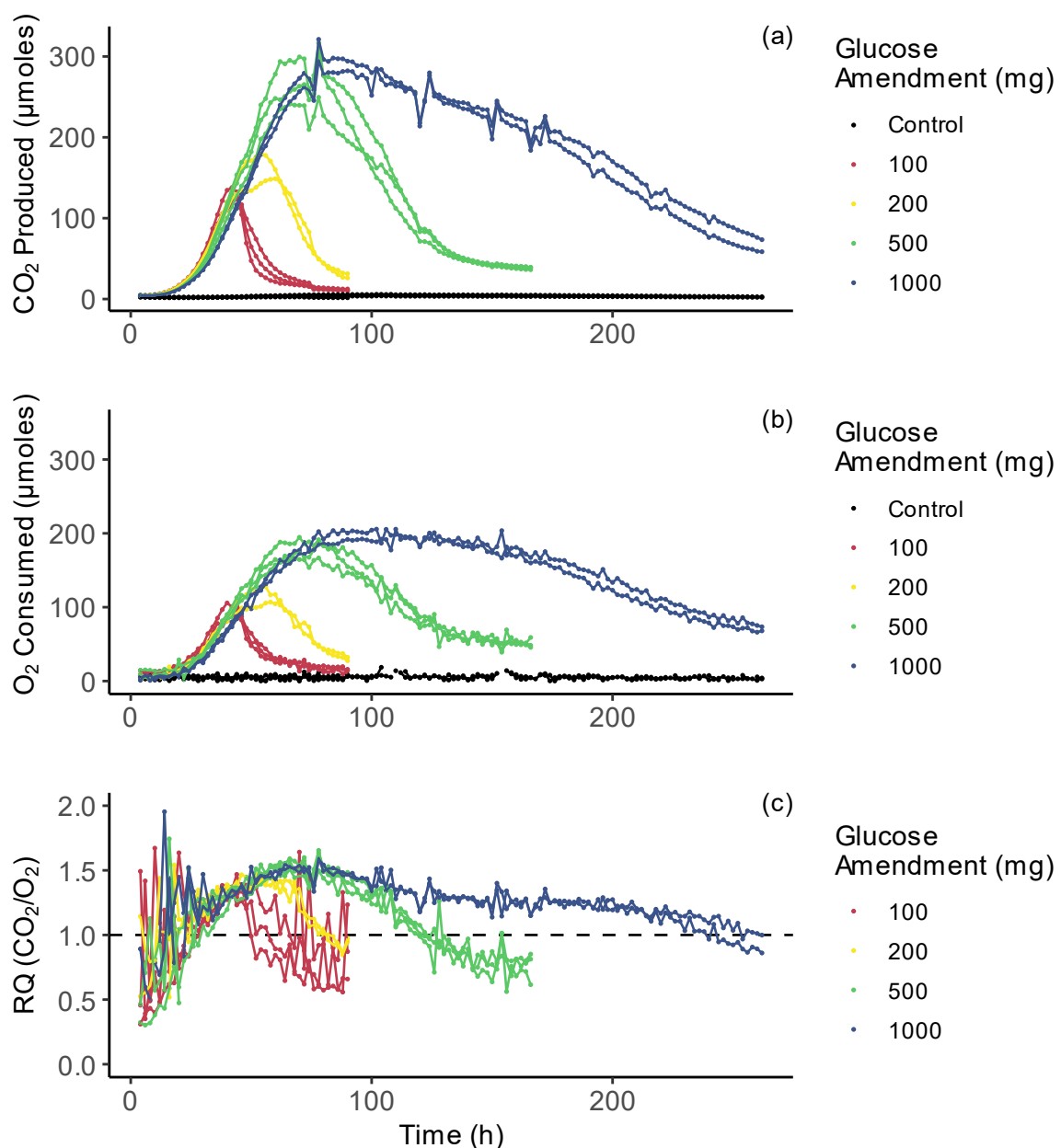

Figure 3: Time series of incubation data. Panel (a) displays $CO_2$ produced in micromoles for each 2 h period. Panel (b) displays $O_2$ consumption in micromoles for each 2 h period. Panel (c) displays Respiratory Quotient (RQ) for each 2 h timepoint, calculated as [$CO_2$ Produced / $O_2$ Consumed].

## 4.2 High Temporal Resolution Carbon Use Efficiency Estimates

Applying the RQ—CUE relationship (Fig. 1) to the incubation data (Fig. 3) allows CUE values to be estimated for each 2 h interval of the experiments. Then, using the $CO_2$ production rate in moles and CUE, biomass production rate in moles-C per 2 h period can be estimated during each time step (Fig. 4b), using the following equation:

$$\text{Biomass Produced} = CO_2 \text{ Produced} / (1 - \text{CUE}) * \text{CUE} \qquad (13)$$

(Fig. 4c) and cumulatively throughout the experiment (Fig. 4d).  It is important to note that once RQ values drop below a value of 1.0, the modeled RQ—CUE relationship for glucose as the sole substrate no longer applies, because 1.0 is the minimal value produced on the metabolism of glucose with this relationship, and lower values would indicate the metabolism of alternative substrates. Further, when RQ values drop below 1.0, this coincides with the point that respiration rates are returning to new basal respiration rates that are elevated over the basal respiration observed in control bottles (Fig. 3). We infer that most if not all available glucose provided in the amendment has been utilized by this point of the incubation. Any further activity is likely driven by metabolism of an alternative substrate, or biomass turnover. Biomass production rates closely resemble respiration rate trends for the incubation. Curves of cumulative biomass produced (Fig. 4d) show all treatments display a sigmoidal shape, which is to be expected as production rates begin low, increase, and then decline back to zero.

Maximum estimated CUE was ~0.56, and the highest values were seen near the beginning of the incubation when RQ values were around 1.9, which may indicate highly efficient growth of small microbial populations, although the small signals produced at the beginning of the incubation may have also been dominated by measurement noise because respiration rates were still low. As respiration rates begin to increase, CUE estimates stabilize at ~0.3, and then continue to increase with respiration rates to ~0.4. A comparison of established methods for measuring CUE (Geyer et al., 2019), shows a wide range of reported CUE values from <0.4 to >0.6 depending on the method of choice. Observations from our incubations sit comfortably within this range, with expected changes over time due to changes in substrate availability (high initially, before gradual depletion).  During respiratory decline, when $CO_2$ production and $O_2$ consumption rates approach new basal conditions, RQ values decline toward 1.0 and CUE estimates fall toward zero. These CUE values, both approaching and departing from, peak activity are plausible because the initial soil should have low standing biomass, and the addition of water and glucose leads to a shift in environmental conditions which are more favorable compared to pre-treatment conditions until substrate depletion occurs and conditions shift back and become less favorable again (Adingo et al., 2021).  As RQ values continue their decline below 1.0 this period may represent a transitionary phase, when the high-lability glucose amendment has been depleted and the microbes begin to turn over and/or target alternate sources of organic carbon. Alternatively, this may represent a period with some final glucose metabolism occurring at an RQ of >1.0 with some metabolism of SOM with an RQ of around 0.7 generating a mixed RQ signal. If this mixed signal is occurring, estimates of biomass production on glucose metabolism may

be slightly too small. Masiello et al. (2008) provides RQ values for other common organic compounds in soils which may serve as these alternate sources during/after decline in RQ. From the list of compounds and their associated RQ's several candidate compounds could satisfy the requirements of our observations; for example, proteins produce RQ's ranging from 0.67-1.01, lignin ranges from 0.88-0.94, and lipids range from 0.68-0.80. Oxidation of any or all of these classes of compounds could explain our observations given that they are basic constituents of plant and microbial biomass and are ubiquitous in soil organic matter.

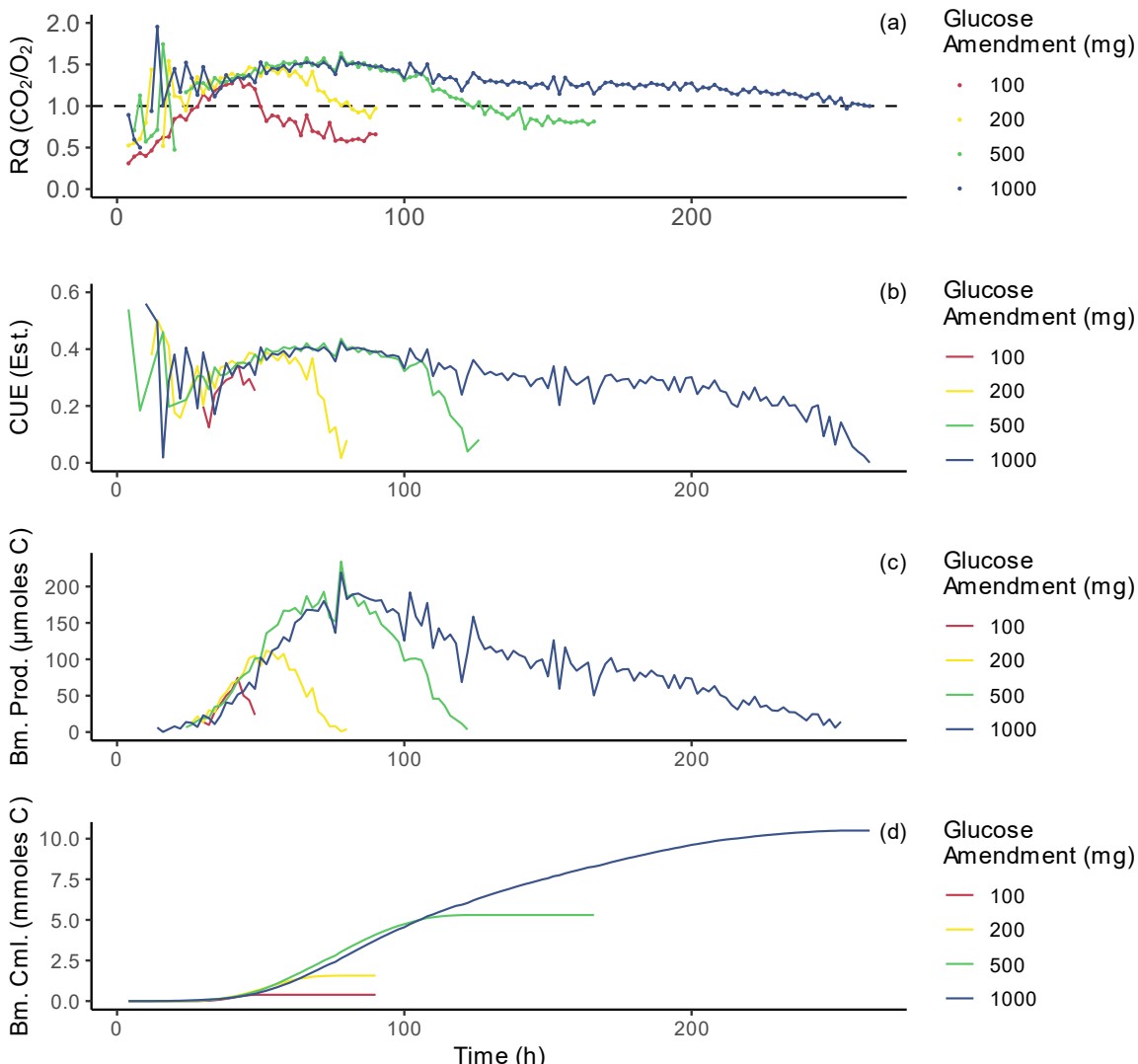

Figure 4: Time series of values calculated from incubation data. All panels present one replicate from each amended treatment (chosen at random) for a visual example of individual sample behavior, additional replicates were hidden from this plot for purposes of clarity. Average values were not presented as temporal offset between replicates would smooth out individual variation. All treatments were carried out through these same calculations. Panel (a) displays RQ over the incubation period,

the same data presented in Fig. 3c are shown here. Panel (b) displays Carbon Use Efficiency estimated for this incubation
using the RQ—CUE relationship presented in Fig. 1. Panel (c) displays micromoles of biomass (carbon) produced at each 2 h
time point for the incubation period. Panel (d) displays the cumulative sum of biomass produced during the incubation in
millimoles carbon.

### 4.3 Understanding the Fate of Amended Carbon

Through the use of the CFE method for microbial biomass carbon measurements along with our gas-based measurements and
determinations of respiration and biomass production, we can construct a carbon budget for each treatment (Fig. 5). Respired
carbon, calculated as the cumulative carbon lost through respiration, shows direct and relatively proportional increase with
amendment size. Salt extractable carbon is presented as (Final $DOC_{unfumigated}$ – Starting $DOC_{unfumigated}$) , which was measured
as part of the CFE biomass calculation and shows a small increase with amendment size. This increase in salt extractable
carbon could be the result of leftover amendment, or from enzymes and other intra/extracellular compounds produced from
the stimulated microbial activity. Measurements of net biomass produced through CFE (Net Biomass produced = Biomass
$_{Amended}$ – Biomass $_{Control}$) on a per bottle basis, show variable, but small increases with amendment. Necromass values, calculated
as (Necromass = RQ $_{Biomass Produced}$ – Net Biomass Produced) show a large increase with amendment size. It is possible that
some overlap between necromass and salt extractable carbon is present. Overall, the sum of these carbon pools nearly equals
the amount of carbon amended to the soil (calculated as 0.4 mg C/mg glucose), as expected for a closed system (Fig. 5), which
provides strong support for our predicted CUE and RQ relationship and further provides some evidence that anaerobic
respiration has not meaningfully taken place in these incubations (Fig.1).

| Treatment | Replication | Amendment (mg C) | Respired (mg C) | Salt Extractable (mg C) | Biomass (mg C) | Necromass (mg C) |
|---|---|---|---|---|---|---|
| A100 | n=3 | 41.7 ± 1.2 | 20.1 ± 0.2 | 4.2 ± 2.3 | 3.7 ± 5.0 | 2.5 ± 5.0 |
| A200 | n=2 | 82.4 ± 1.1 | 39.4 ± 0.3 | 8.5 ± 1.5 | 3.5 ± 1.6 | 14.5 ± 1.6 |
| A500 | n=3 | 200.5 ± 0.5 | 117.2 ± 1.1 | 5.0 ± 1.8 | 4.3 ± 3.3 | 52.3 ± 3.5 |
| A1000 | n=2 | 400.4 ± 2.3 | 263.7 ± 1.9 | 14.5 ± 3.4 | 5.6 ± 1.7 | 118.0 ± 2.6 |

Table 1. Reported mean ± uncertainty of each respective carbon pool in mg C per bottle. Amendment uncertainty is reported
as standard deviation of the replicates, all other uncertainties reported are uncertainty propagated through calculation using
standard deviation of replicates.

Taken together the Biomass, Necromass, and Salt Extractable carbon pools represent carbon that is remaining within the soil
from the amendment after incubation, whereas respired carbon can be considered lost from the soil. With these results, we see
that across the treatments, as amendment size increases a larger portion of the amendment is lost through respiration (~50%
for the 100 mg amendment to ~66% for the 1000 mg amendment), and a smaller fraction of carbon initially amended as glucose

remains in the soil after incubation. However, there are many aspects to this trend that must be considered, such as duration of incubation, long term stability of this necromass, and stoichiometric limitations.

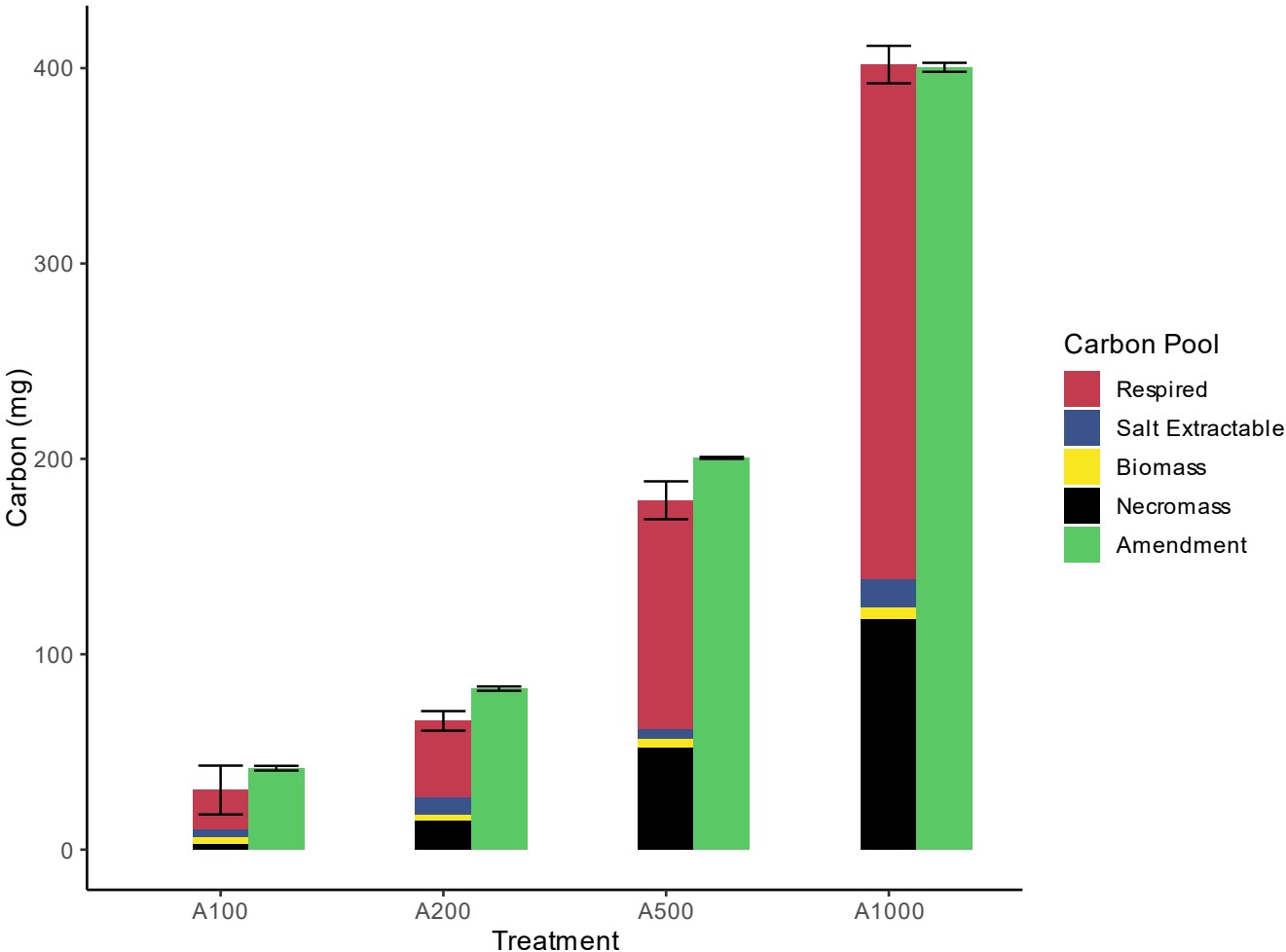


Figure 5. Fate of Amended Carbon. This bar chart shows the respective pools of carbon ascertained through direct measurement, or calculation presented as average with error bars representing uncertainty. Replication varies by treatment (A100: n=3,A200: n=2, A500:n=3,A1000:n=2).

Stoichiometric limitation may be driving the observed increase in the fraction of amended carbon lost via respiration with increasing glucose amendment, considering the carbon amendment was applied without the addition of any other nutrients like N or P. This treatment would drive C:N ratios up, placing the bulk SOM pool in a more carbon enriched state (Table 2), thus would likely drive more waste respiration as other critical nutrients would then be placed in relative limitation (Brown et al., 2022). Further support for this interpretation can be drawn from the slope of the declining RQ values following peak respiratory

activity. During the 100 mg incubation, RQ values declined sharply once respiration slowed, whereas the decline became more gradual with increasing amendment size. Following our model, these decreasing RQ values correspond to decreasing CUE. With larger amendment sizes, there was a longer time interval during which RQ values remained above 1.0 but below the ~1.5 values observed during peak respiration. This period where RQ values are closer to but remain above 1.0 could be explained through a mixture of ongoing glucose fueled metabolism and the onset of microbial necromass turnover, with the latter expected to produce an RQ of ~0.7-0.8. An alternative explanation could be a slower rate of microbial biosynthesis than during peak activity, as increasing nutrient limitation imposes thermodynamic/stoichiometric limitations on biosynthesis and this could be directly reflected in lower measured RQ values as a result of smaller CUE's during the later stages of glucose fueled metabolism.

| Treatment | Replication | C:N | Carbon Weight % | Nitrogen Weight % |
|---|---|---|---|---|
| Initial Soil | n=3 | 20.2 ± 1.5 | 5.12 ± 0.02 | 0.25 ± 0.02 |
| A100 | n=3 | 22.2 ± 2.0 | 5.73 ± 0.24 | 0.26 ± 0.01 |
| A200 | n=2 | 24.9 ± 1.3 | 5.76 ± 0.04 | 0.23 ± 0.01 |
| A500 | n=3 | 25.7 ± 4.2 | 6.17 ± 0.24 | 0.25 ± 0.05 |
| A1000 | n=2 | 28.8 ± 1.7 | 6.65 ± 0.09 | 0.23 ± 0.02 |

Table 2. Measured C:N ratios, carbon weight percent, and nitrogen weight percent of initial soil and incubated treatment soils, reported as mean ± standard deviation.

Recent research shows that after a long (weeks to months) period of incubation, around half of biomass derived carbon may persist within soil as small fragments of cellular envelopes within soil organic matter (SOM) (Kindler et al., 2009; Liang et al., 2019; Miltner et al., 2011). Kästner et al. (2021) highlights a large discrepancy between small quantities of standing live biomass and massive quantities of necromass residue which make up a meaningful portion of SOM. Further, Liang et al. (2019) examined this contribution across ecotypes and found that in temperate forest systems necromass can account for ~30% of soil organic carbon (SOC), though they claim that this lower contribution in temperate forests may be the result of dilution from large continuous inputs of plant material and the lack of tillage. These findings warrant further investigation on the quantification of microbially derived accumulation of SOM, especially through understanding short term microbial metabolism and propagation. The short-term stability of freshly produced necromass in soils remains uncertain. Kästner et al. (2021) describes microbial turnover as a multi-step process where initial cell lysis results in a rapid release of compounds which can quickly stimulate continued biosynthesis, and this cell lysis can be driven through a slower process of starving as substrate availability declines or through more rapid process such as viral activity and microbial grazing (Santos-Medellín et al., 2023). Reflecting on the findings in Fig. 5, we can assess if our data are better explained by substrate depletion and starvation or viral activity and grazing. If substrate depletion and starvation is the dominant driving force behind the formation of necromass, then we might expect greater production of necromass later in the incubation, only once the substrate availability has declined

significantly and RQ values drop below 1.0 and approach ~0.8. In contrast, if viral or grazing activity is the dominant mechanism by which necromass formation occurs, then we would expect a continued formation of necromass relatively in line with the rate of formation of new biomass (Jansson, 2023; Williamson, 2011; Williamson et al., 2005, 2017; Wu et al., 2021). The CFE measurement of biomass carbon occurred immediately after gas measurements ceased, allowing minimal time for further biomass decline. These CFE measurements show that minimal increases in standing biomass production occurred with increasing amendment size, even though very little time passed between the end of the period explained by glucose metabolism (RQ values $\geq 1.0$) and the harvesting for CFE.

This minor increase in standing biomass contrasts strongly with large quantities of total biomass production estimated from the observed RQ values. Taken together, these observations suggest that the rate of new biomass formation during the experiment was similar to the rate of necromass production. Otherwise, we would expect more substantial increases in living biomass once the incubations were stopped. Therefore, viral activity and microbial grazing are considered more suitable explanation, especially considering the treatment of samples as soil was allowed to dry down and were then re-wet (Santos-Medellín et al., 2023; Wu et al., 2021), as recent literature has shown rewetting of dry soil leads to elevated viral activity in a "culling of the victor" strategy. Additionally, considering the shifts in C:N ratios within these samples caused through the amendment of increasing quantities of carbon with no corresponding amendment of nitrogen, we likely drove stoichiometric limitation on the production of new biomass and could have created conditions in this soil which require elevated nutrient mining through strategies such as microbial grazing.

### 4.4 Integrating CUE Over Time

A primary advantage of our methodological approach is the ability to make gas measurements and CUE estimates at a high temporal resolution. Although advantageous moving forward, this complicates direct comparisons to previous studies in which CUE was calculated at lower temporal resolution. Individual estimates during peak respiration, which could best represent the soil microbial community as biomass populations are expected to be at their peak, range from ~0.3 to ~0.4, which compare favorably with the findings of Geyer et al. (2019) as previously stated in section 4.2. Another approach to facilitate comparisons is estimating the net or average CUE over the course of our entire incubation. Integrating our results across time can be accomplished through calculations using data from either Fig 4, or Table 1. In order to understand the most suitable approach to calculating a representative integrated CUE value for each treatment, multiple approaches were carried out. The first approach, which we have termed "1-R.L", is adopted from several previous studies (Adu and Oades, 1978; Anderson et al., 1981; Bremer and Kuikman, 1994; Frey et al., 2001; Shields et al., 1973). This approach calculates the portion of the amendment lost through respiration purely from CO2 measurements, and assumes that all of the glucose amended was taken up by microbes and either respired or used for biomass (Eq. 14). In a modification of this approach, we accounted for the observed increase in Salt Extractable carbon, assuming that this value represents remnant glucose not taken-up by microbes (Eq. 15). We termed an entirely independent approach "RQ$_{Biomass}$CUE", where we consider the gross biomass production calculated from the RQ—CUE relationship (Eq.13) divided by the size of the amendment (Eq.16). We also considered the

possibility that remaining Salt Extractable carbon could represent leftover amendment, and additionally calculated an adjusted calculation in the same manner as with 1-R.L. (Eq.17). Lastly, we integrated our gas-estimated CUE over time using the CUE estimates from Fig. 4b by calculating a weighted average using normalized $CO_2$ production rates from Fig. 3a, termed Gas Weighted RQ—CUE (Eq.18), this method is not adapted from previous works but is a new approach using our new
methodology. Equations for each of these various calculations of CUE are presented:

1-R.L. = 1 - ( Respired / Amended )                                                          (14)

1-R.L.-AA = 1 – ( Respired / (Amended – Salt Extractable ) )                           (15)

$RQ_{Biomass}CUE = ( \sum_T RQ_{Biomass} ) /$ Amendment                                     (16)

$RQ_{Biomass}CUE$ -AA $= ( \sum_T RQ_{Biomass} ) / ($ Amendment – Salt Extractable $)$             (17)

Gas Weighted RQ—CUE $= \dfrac{\sum_{t \in T} CO_2\ Produced(t) * CUE(t)}{\sum_{t \in T} CO_2\ Produced(t)}$                      (18)

where T is the period when RQ > 1.0. Employing these equations provides the resultant integrated CUE calculations across treatments and is displayed in Fig. 6.

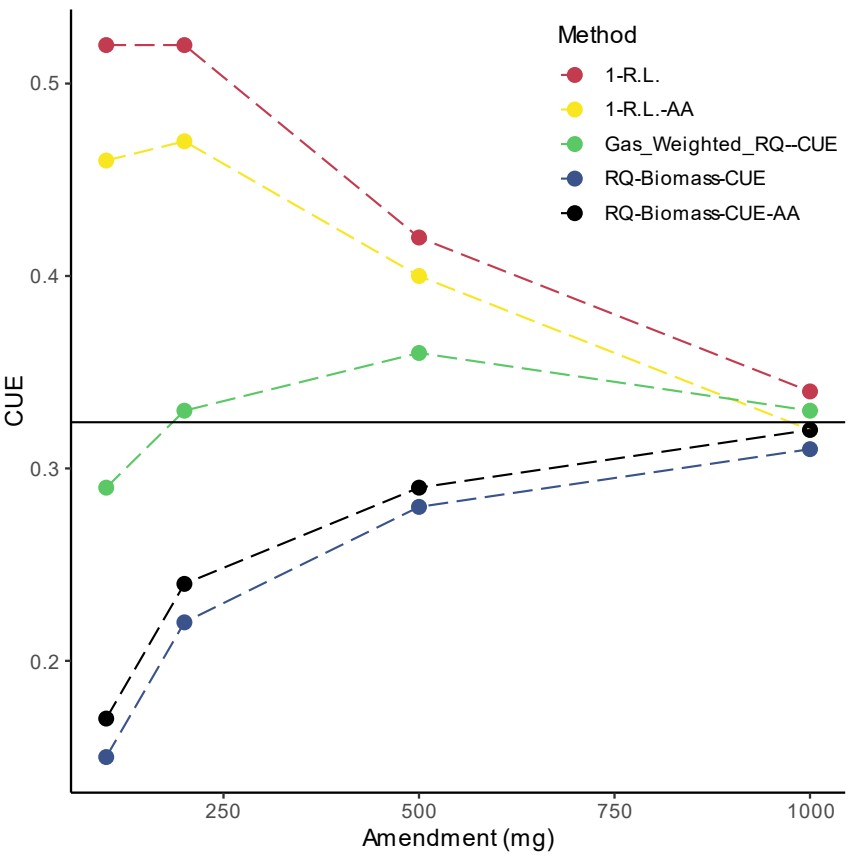

Figure 6. Comparison of Calculated Carbon Use Efficiencies. CUE's calculated from equations 14 – 18 are plotted across soil treatments. Data used to generate these calculated CUE's are sourced from Fig. 3a, Fig. 4b, and Table 1. The solid black line represents the average CUE across calculation methods of the 1000 mg amendment treatment.

Results presented in Fig. 6 show that depending on the CUE calculation methods diverge from each other substantially. There
is a trend whereby the variance among methods decreases with amendment size. Importantly, these values seemingly converge on one CUE ( 0.32 ). This underscores the efforts of recent research (Geyer et al., 2019) in understanding the applicability and implications of using different methodologies to address CUE. It is possible that each of these various methods of calculation tend to capture or consider all the factors which contribute to CUE differently (for instance, one method may inherently consider enzyme production more heavily as a result of the metrics used in calculation). Additionally, as the various calculated
CUE's converge with larger amendments, these subtle inherent differences in calculation may shrink in importance as the magnitude of microbial activity increases. If we consider this convergence on 0.32 as an indication of some true representative CUE for this microbial community, then when considering which method of calculation is most appropriate it would be the Gas Weighted RQ—CUE method as it represents the closest value across amendment sizes to that value of 0.32. Further consideration of these various methods of calculation is required as this value is integrated over time and these various
equations may indeed be capturing different inherent aspect of soil microbial activity.

## 5 Conclusions

A new automated gas sampling apparatus design enabled measurement of high-precision RQ values of glucose-amended and incubated soils at a high temporal resolution (2 h). The non-destructive sampling method allows samples to continuously incubate for a wide range of experimental durations without needing to disturb the incubation chamber. Our results demonstrate
that RQ values observed throughout glucose-stimulated incubations display systematic deviations from the value predicted (1.0) for pure respiration of simple carbohydrates. During peak respiration, RQ values were >1.5, which cannot be explained by a shift to other substrates. Instead, these elevated RQ values during peak activity are best explained by some fraction of the substrate consumed being used to biosynthesize other compounds. Derivation of a stoichiometric relationship between RQ and CUE values enabled measurements of RQ to provide contextual information regarding microbial respiration and biosynthesis.
Not only can this approach provide estimates of CUE at a temporal resolution matching that of RQ measurements, but simultaneous estimates of biomass production can also be calculated by combining this information with $CO_2$ production rate. Importantly, our derived RQ—CUE relation, may be one way forward in real time monitoring of CUE which has proven difficult to measure.
While the representation of all soil microorganisms using just carbon, hydrogen, oxygen, and nitrogen, is a simplification of a
real system, there was close agreement between the estimated carbon pools and size of amendment added (Fig. 5). Further work may address this by incorporation of other potentially important elements, such as phosphorus and sulfur, although we

suspect the simplification will be suitably precise for most applications considering the difference in magnitude of these minor elements relative to carbon, hydrogen, oxygen, and nitrogen in general microbial stoichiometry.

One key advantage of this new method is the ability to monitor soil microbial CUE at a temporal resolution matching that of
measurements of $O_2$ and $CO_2$. This method provides the opportunity to address further questions about carbon stabilization/metabolism on short time scales (hours to days) or intermediate (days to weeks). This method may be applied to address specific questions such as differences in metabolism of various substrates, and the resultant fate of carbon, or this method may be utilized with a few changes to study the effects of oxygen depletion on microbial metabolisms, for example at what concentrations of oxygen do anaerobic respiration become meaningful and important to consider. The implications of
such work could better inform studies which seek to better quantify magnitude and contribution of anaerobic respiration to total soil respiration. Alternatively, this setup could be used to address questions relating to environmental conditions, such as how differences in or transient shifts in soil moisture, temperature, or carbon supply can affect soil microbial populations. The potential applications of this new method, paired with the relative ease of use and minimal oversight required, will enable researchers to address questions that previous methods could not, due to the labor-intensive and time-consuming nature of
traditional laboratory and extraction procedures. Other relationships between CUE and RQ of alternative compounds could be derived for further applications, because the RQ—CUE relationship derived here is only suitable for use when glucose is the primary/dominant substrate undergoing metabolism. Although our study demonstrates that microbial CUE can impact measured RQ values, the RQ values can still act as a rough index of shifts in dominant metabolism, as evidenced by the observed shift to RQ values of ~0.8 after respiration declined significantly. Importantly, these shifts in dominant metabolism
after the amendment of labile substrate are likely driven by the turnover and metabolism of newly produced necromass. However, RQ derived estimates of biomass production were much greater than CFE estimates of standing biomass, suggesting that much of the necromass was not rapidly consumed, although the longer-term stability of this necromass is uncertain. Further consideration of these measurements with a carbon budget reveal that stoichiometric limitation of C:N ratio, could be driving enhanced microbial turnover. Implications of these findings must be considered in the context of environmental conditions,
where heterogeneity of resource availability, and the synergistic mechanisms of a broad microbial community could act to support enhanced carbon stabilization over the long term.

**Data Availability**

The data used in the production of this manuscript are hosted in a Zenodo repository (Smart, 2024).

**Author Contributions**

KES: Carried out experiments and laboratory analyses, analysed data, produced figures, and produced manuscript.

DOB: Developed RQ—CUE relationship, assisted in planning incubation, assistance in interpreting the results, and provided helpful guidance in the production of the manuscript.

CBB: Assisted in planning incubations, interpreting the results, and provided helpful feedback and suggestions in the production of the manuscript.

TMG: Directly advised KES throughout the entire project from conceptualization through the production of the manuscript.

## Competing Interests

The authors declare that they have no conflict of interest.

## Acknowledgements

The authors would like to thank the Kent State University Department of Earth Sciences for funding. This material is also based upon work supported by a GSA graduate student research grant awarded to KES. DOB acknowledges funding from NSF BoCP grant 2224994.

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
