# Peer review of "A new approach to continuous monitoring of carbon use efficiency and biosynthesis in soil microbes from measurement of CO2 and O2"

_EGUsphere, 2024_

## Author Response (AR2)

**Point-By-Point Response Letter**

**Referee #1**

**Comments:**

The reviewed manuscript presents a new approach to the measurement of CUE. Given the importance of CUE for carbon preservation in soils, such a suggestion for a fast and automated approach is intriguing. The calculations connecting RQ to CUE the authors present are straightforward. However, as noted in line 83 there are other controls on RQ including "calcite dissolution/precipitation which can cause a transient decoupling of these two gases, and oxidation of reduced metal species". In addition, anaerobic respiration will cause an increase in the RQ, and oxygen-poor microsites can be found in well-aerated soils. This fact is not noted in the manuscript. This is a major caveat of the suggested approach which requires extensive discussion. It may pull the rug under the entire section 4.3.

**Author Response:** We agree that if meaningful amounts of anaerobic respiration occurred, that could easily be driving elevated RQ values above 1.0. However, there are several reasons that we believe anaerobic respiration is not what is driving the RQ values we measured, and these are listed as follows:

1) Soil headspace gas measurements throughout the incubation were measured, and even at peak respiration oxygen concentrations remained above 19% (these measurements can be found in the zenodo data repository).

2) The soil existed as only a thin layer within our bottles (no more than 1cm thick) and was comprised of particles <2mm in size. Given the strong inhibitory nature of oxygen on anaerobic respiration, we believe this would support abundant aerobic respiration and limit any anaerobic respiration.

3) Anaerobic respiration tends to occur more slowly than aerobic respiration, which further limits its possible impacts on our measurements.

4) The close agreement in the carbon budget presented in figure 5 suggests that aerobic respiration was driving the elevated RQ values we observed.

5) To account for RQ values around 1.5 like we observe, anaerobic respiration rates would need to be significant during peak respiration, when oxygen concentrations were still greater than 19%.

We fully acknowledge that we cannot entirely rule out the presence of any anaerobic respiration from having occurred, however, the data and the controlled incubation conditions suggest that it was likely not significant in these incubations.

**Changes Made To Address Comments:**

Lines 88-92: added "Additionally, the presence of anaerobic respiration can complicate measurements of RQ as microbes utilize alternative terminal electron acceptors to carry out their metabolism which then contributes additional $CO_2$ without the corresponding consumption of

$O_2$. Such processes are particularly important in field studies, when intact soil aggregates allow for the persistence of anaerobic microsites or when soil gases measured are inevitably a result of advection from deeper within the soil profile." – to acknowledge to importance of anaerobic respiration in more traditional respiration measurements.

Lines: 198-201: added "All aspects of this bottle incubation design have been carefully chosen in order to encourage aerobic respiration to be achieved and minimize the possibility for anaerobic respiration to take place, which would impact our measurements and confound our findings." – to underscore the design of the automated sampling apparatus and its intent to promote only aerobic respiration.

I am also missing a comparison of this experiment with experiments measuring CUE using other methods. Is an increase in CUE with added glucose common? Or maybe the peak in RQ at the same time as the peak in respiration indicates that high O2 consumption rates led to local anaerobic conditions?

**Author Response:** To address the second larger concern, the soil pre-treatment as described in line 136, was intentional to place the soil microbes in a stressed state with lower amounts of standing biomass so that when re-wet and amended with glucose, this would stimulate the production of new biomass in effort to measure its effects on RQ. This stimulation of biomass growth would require that bulk soil CUE would be greater than 0. CUE at peak respiration of around 0.4 is generally in line with other works which commonly range from 0.4-0.6. Many of these works, amended glucose to a relatively undisturbed soil in comparison to our soil pre-treatment which makes comparing them precarious. Though it is documented that with changes in environmental conditions like available substrate and soil moisture, changes in CUE do occur.

**Changes Made To Address Comments:**

Lines 317-320: added "A comparison of established methods for measuring CUE (Geyer et al., 2019), shows a wide range of reported CUE values from <0.4 to >0.6 depending on the method of choice. Observations from our incubations sit comfortably within this range, with changes over time after substrate amendment through substrate depletion as may be expected to occur."- to compare our observations with an experiment/meta analysis of commonly used methods.

Lines 321-323: added "These changing values of CUE may be expected to occur as initial soil is expected to have low standing biomass, and the addition of water and glucose leads to a shift in environmental conditions which are more favorable (Adingo et al., 2021)." – to support the idea that CUE should show temporal changes as conditions such as substrate availability changes and soil biomass populations adjust accordingly.

Line 356: added "and further provides some evidence that anaerobic respiration has not meaningfully taken place in these incubations" -to draw support from measured/calculated carbon pools to suggest that no meaningful amount of anaerobic respiration occurred.

Lines 425-470: added "Section 4.4 Integrating CUE Over Time"- to further address the referee comment and produce CUE values which can be compared to other works. To do this we carried out a series of CUE calculations both adapted from previous works and some novel methods

using our data. The results of these calculations are presented in Figure 6 to demonstrate that the same incubation can produce different CUE values based on method of calculation. Further highlighting the findings from (Geyer et. al 2019).

Minor comments:

Line 36: use of "around" followed by a 4-digits-accuracy number is awkward.

Line 36: fixed to "around 1500 Gt…"

Line 82: The sentence there needs re-phrasing. Something like:

other studies observed systematic deviations that were linked to nonmetabolic processes that can affect the soil CO2 and O2 fluxes, and thus noted the ratio of these fluxes as the Apparent Respiratory Quotient (ARQ). Such nonmetabolic processes include…

Line 82: changed to "While some studies report RQ values that resemble substrate-based predictions, other studies observed systematic deviations that were linked to non-metabolic processes which can affect soil $CO_2$ and $O_2$ fluxes,  the ratio of these fluxes have been termed Apparent Respiratory Quotient (ARQ) (Angert et al., 2015). Such nonmetabolic processes include: different diffusion constants of $CO_2$ and $O_2$, calcite dissolution/precipitation, and oxidation of reduced metal species (Angert et al., 2015; Bergel et al., 2017; Gallagher and Breecker, 2020; Hicks Pries et al., 2020; Hodges et al., 2019; Sánchez-Cañete et al., 2018)."

Also, add here the role of anaerobic respiration.

Lines 88-92: added "Additionally, the presence of anaerobic respiration can complicate measurements of RQ as microbes utilize alternative terminal electron acceptors to carry out their metabolism which then contributes additional $CO_2$ without the corresponding consumption of $O_2$. Such processes are particularly important in field studies, when intact soil aggregates allow for the persistence of anaerobic microsites or when soil gases measured are inevitably a result of advection from deeper within the soil profile." -to address the role of anaerobic respiration in traditional respiration studies.

Line 94: Better to just write every 2 hours. High temporal resolution can mean anything from seconds to days.

Line 94- Now Line 99: deleted "at high temporal resolution", now simply " every 2 h".

Line 265: " It is important to note that once RQ values drop below a value 1.0, the modeled RQ—CUE relationship for glucose as the sole substrate no longer applies."

This is confusing: So until RQ=1.0 the model works and then suddenly it is not applicable?  Most likely the same processes causing a RQ<1.0 that the model cannot account for, are already in play before that. So is the model relevant at all? There is a need to expand on this.

Line 265- Now Line 302-305: added ", because 1.0 is the minimal value produced on the metabolism of glucose with this relationship, and lower values would indicate the metabolism of alternative substrates."- to clarify that the relationship is intended to describe the period

dominated by glucose metabolism, and once RQ's decline below 1.0 we can no longer appropriately make that assumption.

Line 299-301: Please rephrase. I could not understand this long sentence.

Lines 299-301-Now Lines 345-357: Rewrote section to improve clarity. "Respired carbon, calculated as the cumulative carbon lost through respiration, shows direct and relatively proportional increase with amendment size. Salt extractable carbon is presented as (Final $DOC_{unfumigated}$ – Starting $DOC_{unfumigated}$) , which was measured as part of the CFE biomass calculation and shows a small increase with amendment size. This increase in salt extractable carbon could be the result of leftover amendment, or from enzymes and other intra/extracellular compounds produced from the stimulated microbial activity. Measurements of net biomass produced through CFE (Net Biomass produced = Biomass Amended – Biomass Control) on a per bottle basis, show variable, but small increases with amendment. Necromass values, calculated as (Necromass = RQ Biomass Produced – Net Biomass Produced) show a large increase with amendment size. It is possible that some overlap between necromass and salt extractable carbon is present."

**Referee #2**

**General Comments:**

The manuscript "A new approach to continuous monitoring of carbon use efficiency and biosynthesis in soil microbes from measurement of CO2 and O2" presents a novel methodology for monitoring microbial CUE through measuring the Respiratory Quotient (RQ). The study is interesting, especially the authors providing a theoretical link between RQ and CUE. The experiments and data analysis are robust. The findings have some useful implications for understanding soil carbon dynamics and microbial metabolism.

I hope the authors can supplement the revised manuscript in the following two aspects:

First, I hope the authors can clarify the specific application prospects of this new method. Currently, there are already various methods for testing CUE, and it is difficult to unify them. This study indeed provides a novel approach. However, I did not read from the text what scientific problems this new method can be used to solve.

**Author Response:** We thank the referee for their thoughtful comments and suggestions.

We believe that the applications of this method and relevant timescale of application are intimately linked questions. The exciting aspect of this approach is the relatively high resolution of these CUE estimates, which could be used to bolster our understanding of fast responses within the microbial community to environmental changes without the time intensive laboratory work associated with other methods for measurement of CUE. The other exciting aspect of this approach is that the metabolism in question can be tailored to answer different types of questions by using the same type of reaction-derivation, this can be applied to many other specific substrates. The second key aspect of this work is that we validate a theoretical link between CUE and measured RQ values. This knowledge will help provide context for field studies of RQ as

well, now with the knowledge that transient shifts may be a result of microbial processes separate from changing substrates.

We believe this method will enable the observation of CUE at resolutions that would be considered logistically challenging with other methods, while the incubations we carry out are done with relative ease after the construction of the apparatus has taken place. We believe the types of scientific questions with which this method would provide strong advantage is those centered on this temporal aspect, for example studying the effects of dynamic environmental conditions on microbial metabolisms and carbon use efficiencies. Further, these incubations could be carried out for extended periods of time (several weeks to months) and provide relatively high resolution on not only CUE when substrate is well constrained, but for example study other processes which are predicted/known to impact measurements of RQ.  With minor modifications, this setup could be used in application to measure soil microbial respiratory behavior as oxygen is used up and conditions begin shifting to anaerobic, in effort to better understand the sensitivity and contribution of anaerobic respiration to bulk soil respiration.

**Changes Made to Address Comment:**

Lines 443-446,Now Lines 555-566: Added "One key advantage of this new method is the ability to monitor soil microbial CUE at a temporal resolution matching that of measurements of $O_2$ and $CO_2$. This method provides the opportunity to address further questions about carbon stabilization/metabolism on short time scales (hours to days) or intermediate (days to weeks). This method may be applied to address specific questions such as differences in metabolism of various substrates, and the resultant fate of carbon, or this method may be utilized with a few changes to study the effects of oxygen depletion on microbial metabolisms, for example at what concentrations of oxygen do anaerobic respiration become meaningful and important to consider. The implications of such work could better inform studies which seek to better quantify magnitude and contribution of anaerobic respiration to total soil respiration. Alternatively, this setup could be used to address questions relating to environmental conditions, such as how differences in  or transient shifts in soil moisture, temperature, or carbon supply can affect soil microbial populations. The potential applications of this new method, paired with the relative ease of use and minimal oversight required, will enable researchers to address questions that previous methods could not, due to the labor-intensive and time-consuming nature of traditional laboratory and extraction procedures."- as guidance or future prospects of how this method could be used to expand our understanding of changes in microbial metabolisms to environmental changes.

Additionally, the authors emphasize that this method can get the temporal dynamics of CUE, as shown in figure 4b. But I think this is also a confusing point of this method: if CUE changes with incubation time, then which time point's value can be used to represent the microbial CUE of this soil sample? Or should it be the average value over a certain period? The answer to this question relates to what scientific problems this method can be used to solve. I hope the authors can supplement their thoughts or suggestions in this regard.

**Author Response:** Our thoughts are that a single CUE that would best represent this period of glucose driven respiration would be a respiration weighted average of CUE over the period when RQ is ≥ 1. We believe this method is the most appropriate representation of CUE of the soil microbes specific to the period of time when glucose is the primary substrate.

**Changes Made to Address Comment:**

The addition of section 4.4 now addresses the difference in CUE value across treatment sizes and highlights that the most discrepancy between calculation methods occurs in smaller amendments, whereas in larger amendments these methods of calculation seem to converge on one value.

Second, this method is based on a mass conservation formula of C, H, O, and N elements to derive the relationship between RQ and CUE. This formula is a simplification of the real ecosystem. I hope the authors can discuss the limitations or deficiencies of this method to help future researchers using this method understand its limitations.

**Author Response:** We acknowledge that only considering C, H, O, and N is indeed a simplification of the real ecosystem and we hope that this work serves as a stepping stone and proof of concept. However, we are assured from Figure 5, that this fairly succinct simplification does effectively capture the process of biosynthesis in a manner which is approachable and compares very favorably with a carbon budget. However, further work could be done to provide an even more descriptive reaction to derive a more complete stoichiometric representation of soil microbes. However, it is also important to remember that other work has shown that microbial stoichiometry can vary with environmental conditions, so a more precise stoichiometry may be more applicable in certain use cases, but not all. Additional elements that could be included in this are phosphorus or sulfur, however their stoichiometric contributions are quite small in comparison to C, H, O, and N.

**Changes Made to Address Comment:**

Lines 484-488: Added "While the representation of all soil microorganisms using just carbon, hydrogen, oxygen, and nitrogen, is a simplification of a real system, there was close agreement between the estimated carbon pools and size of amendment added (Fig. 5). Further work may address this by incorporation of other potentially important elements, such as phosphorus and sulfur, although we suspect the simplification will be suitably precise for most applications considering the difference in magnitude of these minor elements relative to carbon, hydrogen, oxygen, and nitrogen in general microbial stoichiometry."- to provide justification for simplification of the system down to just C, H, O, and N.

**Specific Comments:**

1 Introduction

The introduction is well-written and reads smoothly. However, I noticed multiple formatting errors in the references, e.g., in lines 49, 54, 56, 59, 68, etc.. Please carefully check the formatting of the authors' names in the references.

Multiple lines : fixed in text citations.

2 Connecting Carbon Use Efficiency and Respiratory Quotient

I suggest moving the content from the appendix to the main text. The information in the appendix is essential for understanding how the relationship between CUE and RQ is derived. Additionally, in the appendix, the derivation from step 1 and 2 to step 3 is not straightforward. Could you provide a more detailed derivation process?

Lines 110 118-141, 300: reformatted Equations.

Lines 118 - 141: Moved appendix reaction derivation and added more derivation steps for readers to follow. Also reformatted properly as Equations.

3 Materials and Methods

Line 135: I am curious why glucose was added as a fine solid powder instead of being dissolved in water and then added to the soil. Was there a specific reason for this choice?

Lines 169 - 170 : Added sentence justifying our choice to add glucose as solid fine powder.

4 Results and Discussion

In lines 222, 223, 232, and 233, the figure numbers were missed.

Lines 259-260, 269-270, 298: Added in figure numbers.

Line 265, from the results in figure 4b, it appears that only a small portion of the observations from the 100 mg treatment can be used to calculate CUE. Does this imply that larger doses of glucose should be considered when using this method to measure CUE? I suggest discussing this point.

The addition of section 4.4 now addresses the difference in CUE value across treatment sizes and highlights that the most discrepancy between calculation methods occurs in smaller amendments, whereas in larger amendments these methods of calculation seem to converge on one value.

Figure 4: Why are the results of only one replicate presented instead of the average values of all replicates? If only one replicate result is shown, how did you select which replicate to present from among the several replicates? Please explain.

Line 336 : Added "(chosen at random) for a visual example of individual sample behavior, additional replicates were hidden from this plot for purposes of clarity. Average values were not presented as temporal offset between replicates would smooth out individual variation."